# Application of Machine Learning Algorithms to Predict Uncontrolled Diabetes Using the All of Us Research Program Data

**DOI:** 10.3390/healthcare11081138

**Published:** 2023-04-15

**Authors:** Tadesse M. Abegaz, Muktar Ahmed, Fatimah Sherbeny, Vakaramoko Diaby, Hongmei Chi, Askal Ayalew Ali

**Affiliations:** 1Economic, Social and Administrative Pharmacy (ESAP), College of Pharmacy and Pharmaceutical Sciences, Institute of Public Heath, Florida A&M University, Tallahassee, FL 32307, USA; 2Adelaide Medical School, Faculty of Health and Medical Sciences, The University of Adelaide, Adelaide, SA 5005, Australia; 3College of Pharmacy, University of Florida, Gainesville, FL 32610, USA; 4The Department of Computer and Information Sciences, Florid A&M University, Tallahassee, FL 32307, USA

**Keywords:** All of Us Research Program, machine learning, prediction, uncontrolled diabetes, serum electrolytes

## Abstract

There is a paucity of predictive models for uncontrolled diabetes mellitus. The present study applied different machine learning algorithms on multiple patient characteristics to predict uncontrolled diabetes. Patients with diabetes above the age of 18 from the All of Us Research Program were included. Random forest, extreme gradient boost, logistic regression, and weighted ensemble model algorithms were employed. Patients who had a record of uncontrolled diabetes based on the international classification of diseases code were identified as cases. A set of features including basic demographic, biomarkers and hematological indices were included in the model. The random forest model demonstrated high performance in predicting uncontrolled diabetes, yielding an accuracy of 0.80 (95% CI: 0.79–0.81) as compared to the extreme gradient boost 0.74 (95% CI: 0.73–0.75), the logistic regression 0.64 (95% CI: 0.63–0.65) and the weighted ensemble model 0.77 (95% CI: 0.76–0.79). The maximum area under the receiver characteristics curve value was 0.77 (random forest model), while the minimum value was 0.7 (logistic regression model). Potassium levels, body weight, aspartate aminotransferase, height, and heart rate were important predictors of uncontrolled diabetes. The random forest model demonstrated a high performance in predicting uncontrolled diabetes. Serum electrolytes and physical measurements were important features in predicting uncontrolled diabetes. Machine learning techniques may be used to predict uncontrolled diabetes by incorporating these clinical characteristics.

## 1. Introduction

Diabetes mellitus (DM) is a chronic metabolic disorder characterized by an absolute or relative insulin deficiency [1]. According to the 2021 center for disease control and prevention (CDC) report, about 1 in 10 individuals had DM globally, while 11.3% of the United States (US) population live with DM [2]. Based on the American Diabetes Association (ADA) report, the total cost of diagnosed diabetes was $327 billion in 2017 [3].

DM causes several complications in different vital organs of the body, including the heart, brain, kidney, and eye [4]. One or more diabetes-related complications occur when the blood concentration of glucose is uncontrolled. Uncontrolled diabetes mellitus (UDM) is clinically defined as random blood glucose levels above 180 (mg/dL), or hemoglobin A1c (HbA1c) levels greater than 7.0% [5]. It has cascades of symptoms; however, the occurrence of micro- and macrovascular complications can easily be used to identify UDM. As of 2018, nearly 50% of adults with diabetes had UDM in the US [6].

In order to reduce the magnitude of UDM, different pharmacological and non-pharmacological interventions have been employed [7,8]. The approval of new pharmacological agents and new modes of insulin delivery technology has also improved glycemic control [9,10,11]. Despite these advancements in diabetes management, there is a significant variation in the degree of glycemic control among patients with different characteristics [6]. This fluctuation in glucose control imposes a greater risk of several diabetes-related complications [12,13].

Besides clinical management, early prediction of the glycemic status of diabetes patients could improve the burden of UDM [14]. In the past, few attempts have been made to predict UDM using routine glycemic status measures such as fasting plasma glucose, hemoglobin A1c, and oral glucose tolerance tests. The routine glycemic tests might not be convenient as they require overnight fasting. In addition, previous approaches have yet to include larger biological marker data. Particularly, the role of biological biomarkers such as serum electrolytes and hematological indices in predicting UDM using machine learning (ML) algorithms remains partially understood, despite their interaction with glycemic status [15].

ML algorithms are the state-of-the-art techniques implemented to predict various outcomes in medicine, including the new onset of diabetes, antidiabetic drug safety and utilization, and diabetes complications [16,17,18,19]. Nevertheless, a small number of studies implemented ML methods to predict glycemic control in diabetes patients using selected features. For instance, Del Parigi, A. et al. (2019), [20] Lee, S. W. et al. (2019), and Basu, S. et al. (2012) [14,21] developed various ML models to predict glycemic control. These studies reported the role of ML techniques using limited clinical features that rely on routine glucose monitoring parameters such as fasting plasma glucose and HbA1C. However, there was a lack of consideration of several patient-level inputs such as serum electrolytes, anthropometric measures, blood indices, lipid profiles, and organ function tests in their predictive model development. 

In addition, physical measurements including weight and height might indicate the presence of obesity, which usually occurs with hyperglycemia [22]. On the other hand, blood indices such as erythrocyte counts influence the hemoglobin level, affecting HbA1C [23]. The correlation between one or more of these features with glycemic status can provide an alternative method of monitoring UDM, if an appropriate predictive model is developed using data derived from a representative sample. The present study used the All of Us (AoU) research program, a representative sample of an ethnically diverse US population which provides large sample size [24].

The prediction of UDM using new features could be a cost-effective strategy for glucose monitoring compared to standard glucose measures. In addition, the implementation of a new predictive model with more patient characteristics would help decrease diabetes-related complications and improve the quality of life of patients with diabetes through the screening of at-risk individuals. In light of this, our study employed multiple patient characteristics to efficiently predict UDM using supervised machine learning. The study highlighted the significance of physiologic biomarkers and patient characteristics to predict UDM in the absence of regular glycemic status measurements. It is important to note that these clinical features are crucial in predicting UDM and identifying patients who require further management. 

By utilizing a predictive model that incorporates various patient characteristics, healthcare providers can identify patients with UDM at an earlier stage, allowing for prompt intervention and management. Therefore, this paper aims to explore patient characteristics and biomarkers in predicting UDM by applying different ML algorithms. 

## 2. Materials and Methods

This section is composed of data sources, population characteristics, data processing, model selection, model development, feature importance and performance metrics. The data source section describes the nature of the data and how it was accessed. Population characteristics detail the demographic and clinical characteristics of the study participants. The data processing procedure indicates the steps taken, including merging, feature engineering, handling sample imbalance, preventing data leakage, and variable standardization. Model selection and development refer to the type of ML algorithms that have been employed in the study and the optimization of ML algorithms (i.e., hyperparameter tuning) while the performance metric indicates measures such as AROC, accuracy, precision, and recall, which show the predictive ability of the ML algorithms. 

### 2.1. Source of Data 

The current study was conducted using data from the AoU Research Program. The survey details and data collection method can be found here (https://allofus.nih.gov/ (accessed on 8 April 2023)) [25]. In brief, the AoU database is a longitudinal database collected in multiple rounds. The components of the data include health questionnaires, electronic health records (EHRs), physical measurements, digital health technology, and biospecimens. Since 2018, the research program has enrolled a diverse group of at least 315,000 persons in the US aged 18 or older from a network of recruitment sites. More than 80% of the participants are from groups that have been historically underrepresented in biomedical research, including 49% from non-white races [24,26]. The data were accessed using the AoU Research Workbench, a cloud-based platform that enables approved researchers to access and analyze the data. Participants provided informed consent to participate and provided authorization to share EHR data with the Data and Research Center of the AoU research program. The physical measurement and biospecimen collection occur at the initial enrollment visit, following consent and completion of the survey. 

### 2.2. Population Characteristics 

The AoU researcher workbench was used to characterize our study population. Both type 1 diabetes mellitus (T1DM) and type 2 diabetes mellitus (T2DM) patients aged above 18 years who were on glucose-lowering medications were included. Pregnant, children and patients with an incomplete record of diabetes control status and other comorbidities were excluded. The AoU Researcher Workbench provides a built-in tool for selecting participants (Cohort Builder), creating a dataset for analysis (Data Builder), and creating a Workspace with R and Python notebooks to analyze the data. The total population is composed of 33, 826 diabetes patients. Of these, about 20% of them had UDM. Around 43.43% of participants were female, and forty-three percent of individuals were non-Hispanic whites (Appendix A). 

### 2.3. Data Processing and Mining 

The data-processing procedure described here was performed using R version 4.0.2 (the R Project for Statistical Computing) R software [27]. The data collected on different variables were merged using patients’ identifiers, whereas data collected on similar variables from different subjects were appended. Standardization of continuous variables was undertaken. Information on glycemic control status was documented as “diabetes with hyperglycemia” in the AoU research database, based on the international classification of disease (ICD). Controlled DM was labelled = 0, while UDM was labelled = 1. All relevant features with a missing value of less than 25% were included. 

Different steps of feature engineering were undertaken, including outlier detection, standardization of continuous variables, imputation to handle missing values and creating one-hot matrix/encoding. The outlier data points were detected and removed using domain knowledge and data visualization. One-hot encoding was used to convert categorical variables into new variables that took on values of 0 and 1 to represent the categorical values. This format allows categorical variables to be used easily by machine learning algorithms. In addition, the data were imputed to handle missing values provided that the number of missing values of a given feature was less than 25%. 

The class imbalance was handled using random over-sampling examples (ROSE), a smoothed bootstrap-based technique [28]. Handling imbalanced data involves preventing the effect of imbalanced values of the outcome variable on model performance. In our study, the dataset contained 80% of patients with controlled DM and 20% patients with UDM, which gives an imbalance ratio of 1:4. Classes that make up a large proportion of the dataset (controlled DM) are called majority classes. Those that make up a smaller proportion (UDM) are minority classes. In order to handle the imbalance, a random sampling technique was applied using the random over sampling examples (ROSE) package in R to resample the training dataset so that nearly equal proportions of UDM and controlled are created. The current study utilized three main approaches to random sampling. The first approach is called random under-sampling, which deletes examples from the majority class. The second approach, random over-sampling, duplicates examples from the minority class. A third approach was also tested by combining both random over-sampling and under-sampling techniques. Eventually, the sampling approach that gave the highest model performance was selected. 

In order to prevent data leakage, several procedures were conducted. These include dropping duplicate observation in the training and testing set and removing features that were highly correlated with the outcome variables, such as the fasting plasma glucose (FPG) and hemoglobin A1C (HbA1C). A multicollinearity test was performed between features using the variance inflation factor (VIF) test, and variables that had a VIF value of greater than 5 were removed.

The diagram in Figure 1 illustrates the different subsections of the Materials and Methods sections. The Methods section mainly comprises the following categories: the source of the data and population, data processing/mining, model selection and rationale, and model development and optimization steps, as described below (See Figure 1). 

### 2.4. Model Selection and Rationale 

A supervised ML algorithm was applied to predict UDM and to identify important features. Supervised learning is an ML method in which an algorithm is fed with data that contain observations and a label. This predictive model was based on extreme gradient boost (XGBoost), random forest (RF), logistic regression (LR), and a weighted ensemble model (WEM). The XGBoost and RF models are the most common supervised learning algorithms used for classification and regression problems by assuming nonlinear relationships between variables. These ML algorithms have a high prediction accuracy and can handle the multicollinearity of features. The XGBoost and RF are ensemble models based on decision trees [29,30]. A WEM was also established by combining the predictive probabilities of the three individual models to obtain a robust model with better performance from weak learners. The three individual models (RF, XGBoost and LR models) were represented by (*K* = 1, 2, and 3). Initially, a weight parameter *(Wk)* was calculated by using the area under the receiver operating curve *(AROC)* values of each model. We can assume *AROCk* is the *AROC* of model *k* (Equation (1)). An example was provided below on how *Wk* was calculated using individual *AROC* values reported in Section 3.1.
(1)$$Wk=\frac{AROCk^{2}}{\sum_{k=1}^{3}AROCk^{2}}$$

For instance, *Wk* for RF = $Wk=\frac{{\left(0.77\right)}^{2}}{{\left(0.77\right)}^{2}+{\left(0.75\right)}^{2}+{\left(0.70\right)}^{2}}=$ 0.36

Then, the prediction probabilities were multiplied by the weight of each model. The result was summed up to give the predictive probability of the ensemble model (Equation (2)). Each patient was then classified according to weighted probability. *Pk* represents the classification probability of each model, and *Pw* is the predictive probability of the ensemble model.
(2)$$Pw=\sum_{k=1}^{3}WkPk$$

### 2.5. Model Development and Optimization of Parameters 

The training and the testing datasets were created by randomly splitting the data into an 80% versus 20% split ratio. The training dataset (80%) was utilized for training the model, while 20% of the dataset was left for testing. A cross-validation technique was used to estimate how well our model generalizes to new, unseen data. In the cross-validation approach, the training data were partitioned into multiple subsets, and the model was trained on a portion of the data while evaluating its performance on the remaining portion. This process was repeated multiple times and the results were averaged to obtain an estimate of how well the model is likely to perform on new, unseen data. During cross-validation, the model was assessed for underfitting and overfitting the data.

After model training, model validation was carried out on the testing set (a new example dataset that was not used to build the model). Necessary adjustment was carried out to enhance model’s accuracy/performance during validation phase. The hyperparameters were tuned to obtain a robust ML model with improved performance [31,32]. Hyperparameter tuning consists of finding a set of optimal hyperparameter values for a learning algorithm that maximizes the model’s performance and reduces errors. The tuneRF tool in the RF Package was used for tuning the RF model. The RF model was tuned by using common hyperparameters including the number of trees used at the tuning step (ntree), the number of variables randomly sampled as candidates at each split (mtry), and node size. The optimization of the model started with the default value of hyperparameters before searching until the optimal hyperparameter value that gives a smaller out-of-bag (OOB) error estimate was obtained. The OOB error decreased as the number of trees increased, which suggested the improved performance of the RF model (Appendix A). 

For the XGBoost model, learning curves were used for model tuning, while Mlogloss of the training and testing set was applied as an evaluation metric. A learning curve is a plot that shows number of iterations in the x-axis and the mlogloss on the y-axis (Appendix A). The smaller the mlogloss, the higher the learning or improvement in the model’s performance. Two learning curves were drawn on the training and testing sets, respectively. The curves were then used to diagnose under or overfitting of the XGBoost model. Accordingly, further hyperparameter tuning was carried out until the model became a good fit. This was achieved by tuning different hyperparameters including but not limited to the learning rate (eta), number of rounds (nrounds) and maximum tree depth (max.depth). 

### 2.6. Feature Importance 

A total of 54 features were included in the model after dropping all variables with more than 25% missing values. This includes six demographic features, eight medications, 11 hematologic indices, 18 biomarkers and 11 physical measurements as relevant biomarkers for the prediction of UDM. The variable importance indicates the contribution of each feature to predicting the outcome. The variable importance was derived from the best predictive model among all the examined models. The most important features were identified using the mean decrease in accuracy score [33]. Mean decrease accuracy is a variable importance measure that expresses how much accuracy the model loses by excluding each variable. The more the accuracy drops in the absence of a given variable, the more important the variable is for accurate classification. The unit is expressed as a percentage (%). 

### 2.7. Performance Metrics 

The performance of each model was evaluated using the following performance metrics: precision, recall, classification accuracy, F1 score, and area under the ROC curve (AROC). F1 score is an ML evaluation metric that measures model performance. F1 was calculated using a formula that incorporated precision and recall, $F1=\frac{2*\mathrm{precion}*\mathrm{recall}}{\mathrm{precision}+\mathrm{recall}}$. The numerator is product of precision and recall times by two, while the denominator is the sum of precision and recall. The value of F1 score ranges from zero to a hundred and is expressed as a percentage. Higher F1 values indicate the high performance of the model. Precision and recall are the alternative measures of ML performance. Precision measures the accuracy of positive prediction, whereas recall measures the percentage of data samples that a machine learning model correctly identifies [34]. Precision and recall were calculated using the following formula:$$\mathrm{Precision}=\frac{\mathrm{True}\;\mathrm{Positve}}{\mathrm{True}\;\mathrm{Positive}\;+\mathrm{False}\;\mathrm{Positive}}\;\mathrm{and}\;\mathrm{Recall}=\frac{\mathrm{True}\;\mathrm{Postive}\;}{\mathrm{True}\;\mathrm{Positive}\;+\mathrm{False}\;\mathrm{Negative}}.$$

A true positive is an outcome in which the model correctly predicts the positive class, while true negative is an outcome in which the model correctly predicts the negative class. A false positive is an outcome in which the model incorrectly predicts the positive class, and a false negative is an outcome in which the model incorrectly predicts the negative class [35].

## 3. Results

The main findings of the study summarized the classification performance of models in predicting UDM. They also outlined the important features that predict UDM, including electrolytes, blood indices and physical measurements, based on the mean decrease accuracy of the RF model. 

### 3.1. Classification Performance of Models to Predict UDM 

Among the ML models, RF has the highest AROC and a prediction accuracy of 0.77 and 0.80 (95% CI: 0.79–0.81), respectively. The WEM model had 0.77 AROC and a prediction accuracy of 0.77 (0.76–0.79), while the XGBoost model attained a comparable AROC of 0.75 and a prediction accuracy of 0.74 (0.73–0.75). The LR model had an AROC of 0.70 with a classification accuracy of 0.64 (0.63–0.65), which is lower than other models (Appendix A, Figure 2A). The performance of the ML models in predicting UDM separately for females and males was further examined. There was no marked difference in accuracy for the RF model across both genders. In contrast, the prediction accuracy of the XGBoost increased in males versus females. The LR model demonstrated poor performance in both genders (Figure 2B,C). AROC curves were drawn for each model, which provided additional information on the performance of the ML models (Figure 3, Appendix A). 

### 3.2. Feature Importance 

According to the mean decrease accuracy score of the RF model, serum electrolytes, physical measurements, liver enzymes, and vital signs were found to be significant predictors of UDM; notably, serum potassium concentration, body weight, aspartate aminotransferase (AST), heart rate, and systolic blood pressure were significant predictors of UDM when the analyses was performed without classifying the dataset in terms of gender (Figure 4). It was noted that variability in features’ importance was observed between males and females to some extent. In female DM patients, height was a more important feature than body weight in predicting UDM. In addition, blood indices, including erythrocytes and leukocyte counts, were found to be more significant predictors. In contrast, body weight was the primary determinant of UDM in males (Appendix A). 

The X-axis represents the mean decrease in model accuracy, which is expressed as a percentage (%). 

Abbreviations: AST: aspartate aminotransferase, ALT: alanine transaminase, ALT: alkaline phosphatase (ALP), DBP: diastolic blood pressure, MCH: mean corpuscular hemoglobin MCHC: mean corpuscular hemoglobin concentration, MCV: mean corpuscular volume, SBP: systolic blood pressure. 

## 4. Discussion

The present study applied a data-driven ML approach to predict UDM in a comprehensive dataset from the AoU Research Program. The study also explored important features that could improve prediction of UDM. It was found that the RF model demonstrated high accuracy in predicting UDM as compared to the XGBoost, LR, and WEM models. The overall performance of the predictive ML models ranged from 80 percent (RF model) to 64 percent (LR model).

Few ML models have been employed previously to predict poor glycemic control in a different patient population. For instance, Basu S et al. (2019) developed RF and gradient-boosting algorithms, and reported that the RF model was superior in predicting UDM over other models [14]. This study corroborates the findings of a great deal of the previous work, in which the RF model demonstrated a higher prediction accuracy. For example, an ML-based study on T2DM patients admitted to Sichuan Provincial People’s Hospital reported a classification accuracy of 0.72 to 0.76 using neural networks [36], which is a relatively lower accuracy than our RF model. This variation could be attributed to the implementation of different ML models and the inclusion of different features as the main risk factors of UDM in the previous study [36]. Our prediction finding is contrary to that of Motaib, I. et al. (2022) who predicted poor glycemic control during Ramadan using the extra tree classifier (accuracy = 0.87, AUC = 0.87), which is relatively higher in prediction accuracy as compared with our study findings [37]. This rather contradictory result might be due to the inclusion of baseline caloric intake as an important factor in their study. In addition, Tao, X. et al. (2022) predicted glycemic control from a total of 375,723 cases of DM patients with the random forest model (AROC = 0.97), which exhibits higher classification efficiency than our models. This result may be explained by the difference in the sample size, wherein they included a large sample size, and another possible explanation for this might be the inclusion of the baseline glucose level as a predictor variable [38].

In the current study, various factors have been explored as predictors of UDM. Importantly, potassium was the most influential predictor, followed by weight, AST, height, and heart rate, when both female and male populations were included in the model. A possible explanation for these results may be the cellular level involvement of potassium. Potassium is an electrolyte that participates in the metabolism of glucose through various pathways. For instance, potassium affects glucose metabolism through its impact on the secretion of physiological insulin [39]. When there is a low potassium level, the ATP-sensitive potassium (KATP) channel in pancreatic beta-cells cannot stimulate potassium-dependent insulin release, resulting in higher blood sugar levels [39]. Another possible explanation is that potassium can also interact with the renin–angiotensin–aldosterone system (RAAS), which affects glucose tolerance [40]. The activation of the renin–angiotensin–aldosterone system reduces potassium concentrations in the body, thereby inhibiting insulin release and increasing glucose intolerance [41]. In accordance with the present results, previous studies have demonstrated that potassium has an important role in controlling blood glucose level. For example, the Jackson heart study reported that potassium concentrations were associated with hyperglycemia due to its influence on aldosterone levels [42].

Furthermore, one of the microvascular complications of UDM is kidney damage, which causes potassium accumulation. Potassium accumulation might result in a rapid heart rate [43,44]. This is also reported as a predictor of UDM in our study. The disposition of potassium in the body also affects sodium transport through Na+-K+-ATPase, [45] which in turn affects glucose reabsorption in the renal tubule through the sodium–glucose co-transporter [46]. In summary, these biochemical interactions between potassium and other biomarkers indicate that potassium measurement could be pivotal in predicting glycemic status [47]. 

In the present study, physical measurements such as weight and height were also the other important features in the prediction of UDM. These parameters are used to determine body mass index (BMI), which was found to be correlated with hyperglycemia and has been used to predict glycemic status [48,49]. Being overweight and having glucose intolerance are the two main components of metabolic disorders that can coincide [50]. Moreover, overweight results in fat accumulation in the liver and the production of inflammatory cytokines that impair insulin signaling and tissue sensitivity to insulin action, which causes a flaw in glucose metabolism [51,52]. This physiologic relationship between physical measurements and glucose could help to predict glycemic status using these parameters. 

Our study also reported that liver biomarkers, including AST, can predict UDM. AST is a liver enzyme that participates in gluconeogenesis in the liver and kidney [53]. AST mediates glucose synthesis in the liver from amino acids, which promotes hyperglycemia [54]. A study conducted to determine the association between hyperglycemia and liver enzymes has found that the rate of hyperglycemia increases as the level of AST elevates [55]. Sheng X et al. (2018) investigated that insulin resistance was higher in individuals with a high concentration of liver enzymes, which might lead to UDM [56]. A longitudinal study on the Korean population pointed out that serum levels of liver enzymes were associated with incident hyperglycemia [57]. Insulin resistance and pancreatic damage were also correlated with liver aminotransferases [58]. Thus, the inclusion of liver enzymes in ML models could improve the prediction of UDM. 

Furthermore, the current study also revealed that blood indices such as erythrocytes and leukocytes as significant predictors of UDM, especially in females. The reason for this is not clear, but it may have something to do with the blood indices. When there is hyperglycemia, the morphological structure and physiological functions of erythrocytes are severely affected [59]. This finding broadly supports the work of other studies in this area linking blood glucose level with hematological indices. For instance, a comparative observational study reported a statistically significant decline in the erythrocyte count of diabetes patients [60]. This suggests that blood indices might be correlated with glycemic status. In addition, erythrocytes can also affect hemoglobin levels. Hemoglobin is a protein in the erythrocyte that helps transport oxygen in the blood. Its amount increases with the number of erythrocytes [61]. Hemoglobin binds with blood glucose to form glycohemoglobin, one of the standard measures of glycemic status in diabetes patients. Therefore, the measurement of erythrocyte indices might reflect the level of glycohemoglobin in the blood, or a patient’s glycemic status [23].

### Strengths and Limitations of the Study

In general, the current study predicted UDM with state-of-the-art ML algorithms using large datasets incorporating data from the underrepresented population. The application of ML algorithms improves the prediction of UDM over statistical models. The findings could be generalized for diverse groups of diabetes patients in the US. The study also investigated different biochemical features that were not reported in previous studies that can predict UDM in absence of the routine glucose monitoring parameters such as fasting plasma glucose and hemoglobin A1C. These features will enable the simultaneous monitoring of electrolytes, metabolic syndrome, and blood disorders along with UDM. This might help to make use of samples more efficient, decrease test invasiveness, and decrease laboratory costs. However, this study depends on the AoU data that contain only adults 18 and older that are currently residing in the U.S. Hence, the research findings may not be generalized for diabetes patients who reside outside of the US. The present study also did not capture social determinants, as these variables were incorporated into the AoU research program only recently. In order to overcome these limitations, prediction of glycemic control using continuous glucose monitoring data might also help apart from the AoU dataset. More recent datasets such as the ShanghaiT1DM and Shanghai T2DM are design for data-driven machine learning to predict glycemic control in DM patients [62]. In addition, other studies have also reported the role of reinforcement learning in predicting blood glucose control; these complement the current study [63]. Further to this, for a more comprehensive approach in predicting UDM and to potentially improve the management of diabetes and prevent complications associated with UDM, future studies are needed to investigate the impact of additional features, such as social determinants, dietary and lifestyle factors, on the accuracy of our models. Furthermore, a cost-effectiveness analysis can be performed to determine the practicality of implementing these machine learning models in clinical settings.

## 5. Conclusions

In summary, this study assessed the effectiveness of machine learning algorithms in predicting UDM. The findings showed that random forest (RF)-based models were more efficient in predicting UDM compared to other machine learning algorithms. Moreover, the study identified that serum potassium concentrations, blood indices, and physical measurements were important features in predicting UDM. We noted that variability in features’ importance was observed between males and females to some extent. In females, height was a more important feature in predicting UDM. In contrast, body weight was the primary predictor of UDM in males. Our findings have significant implications for clinical practice, as these parameters could be incorporated into clinical guidelines at minimal additional cost in order to monitor glycemic status in cases wherein regular monitoring techniques are not available. However, to fully integrate these findings into regular monitoring techniques, further investigation is needed to determine their cost-effectiveness and the practicality of their implementation in clinical settings. Such research would help to clarify the potential benefits and challenges of using machine learning algorithms to monitor glycemic status, and potentially improve the management of UDM and prevent complications associated with uncontrolled diabetes. Therefore, more detailed research is warranted to further explore these issues.

## Figures and Tables

**Figure 1 healthcare-11-01138-f001:**
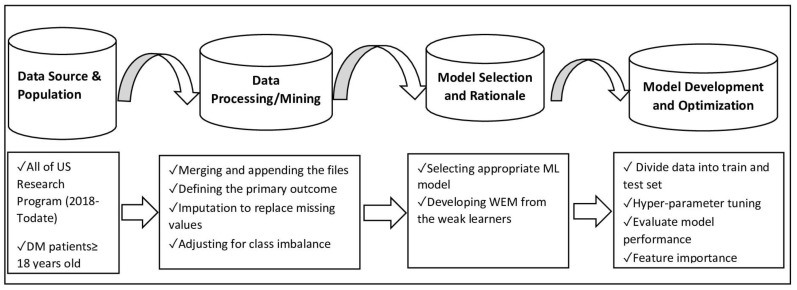
Flow diagram for data processing and machine learning-based model development steps in UDM using AoU Research Program data.

**Figure 2 healthcare-11-01138-f002:**
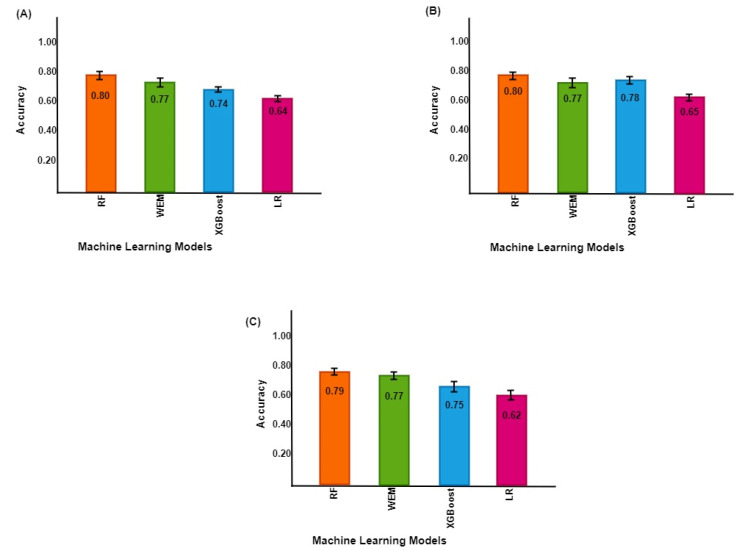
Uncontrolled DM prediction accuracy across models of machine learning in the AoU Research Program. The panel in the left top corner (**A**) shows the prediction accuracy of ML models for both genders. The right top corner (**B**) is the prediction accuracy for females, and the right lower panel (**C**) is the prediction accuracy for males.

**Figure 3 healthcare-11-01138-f003:**
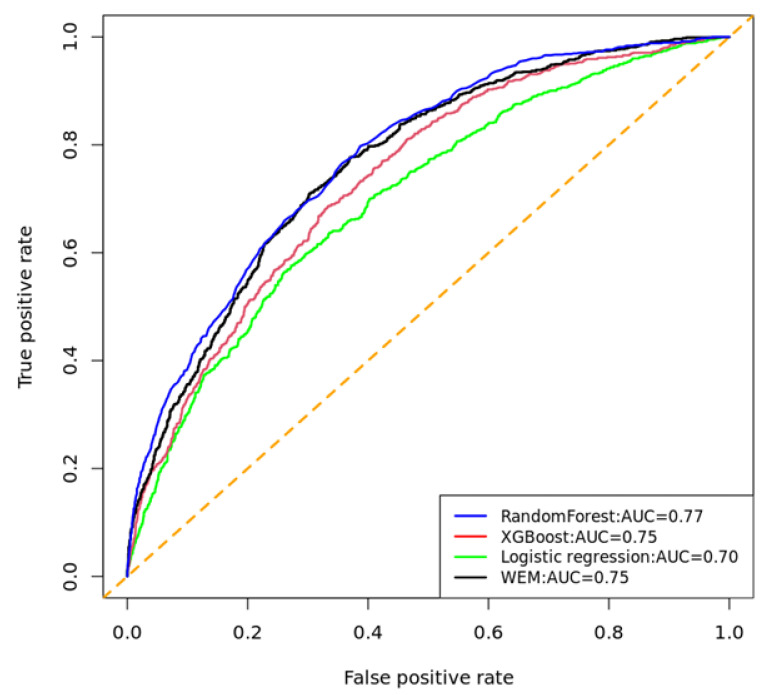
Comparison of receiver operating characteristic (ROC) curves of the ML models implemented to predict UDM in AoU, 2023. AUC: area under the curve. The diagonal line dotted in orange indicates that true positive rate is equal to false positive rate.

**Figure 4 healthcare-11-01138-f004:**
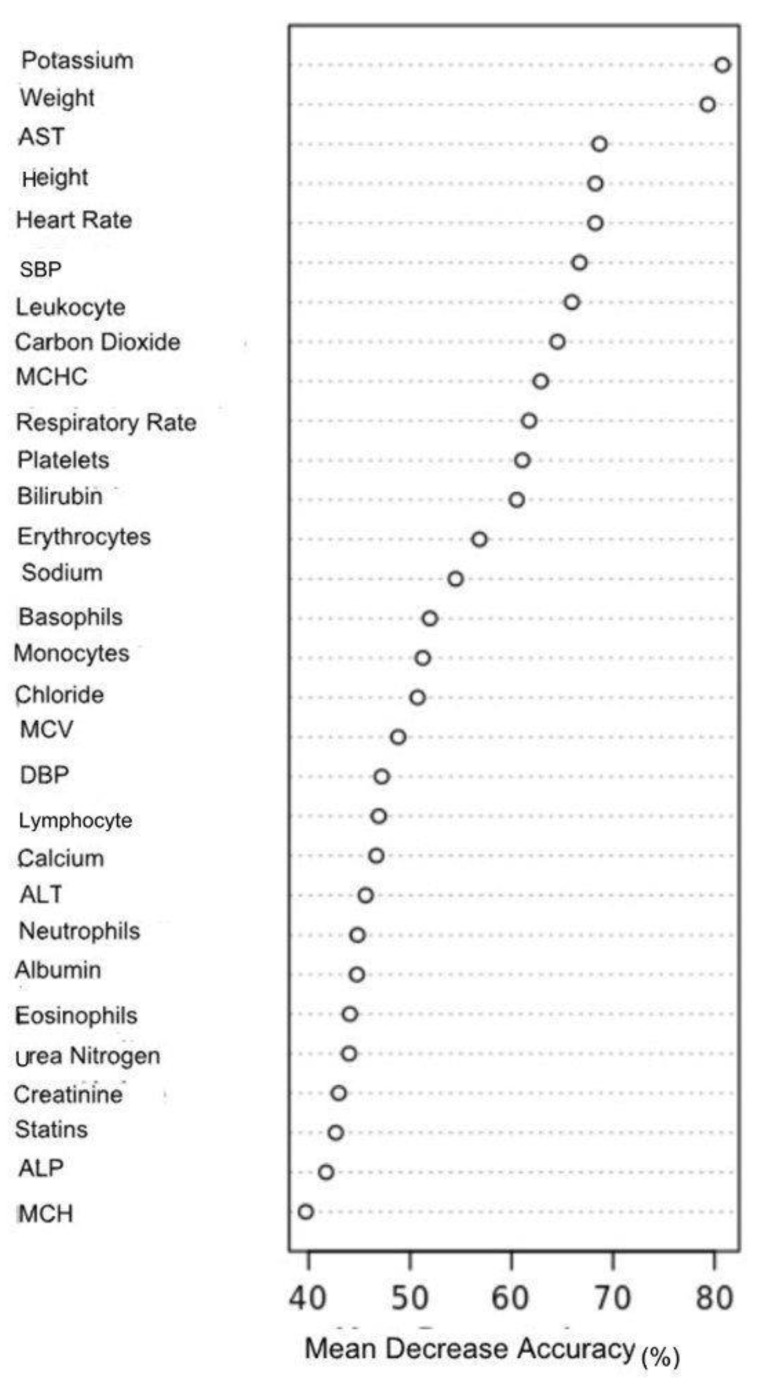
Feature importance for the prediction of UDM in AoU Research Program 2023.

## Data Availability

The authors confirm that the data supporting the findings of this study are available within the article and its Appendix A.

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
