# Peer review of "Application of Machine Learning Algorithms to Predict Uncontrolled Diabetes Using the All of Us Research Program Data"

_healthcare, 2023, doi:10.3390/healthcare11081138_

Round 1
Reviewer 1 Report
The manuscript by Abegze TM et al investigated the application of ML in predicting uncontrolled diabetes disease. The research has clinical and economical importance due to the fact that diabetes affects a significant amount of the population in the US and comes with high economic costs to get diagnosed. The manuscript has flaws that must be fixed and improved:
· There’s no mention of how the models are validated.
· It’s unclear whether the results shown are on the training datasets or testing datasets. The model performances must be shown on both datasets to evaluate if the models overfit or underfit.
· There’s no mention of how the data leakage was prevented in the models.
· Figure 4: what’s the unit for x-axis.
· There’s no mention of whether any features were engineered to improve the model performance.
· There’s no mention of how the models were tuned and optimized.
· It’s difficult to justify what the data looks like. Please include the standard deviation for continuous variables in eTable1.
Author Response
Reviewer 1
Dear Reviewer,
Thank you so much for your valuable comments. We have responded to your comments point by point as follows.
Comments and Suggestions for Authors
- There’s no mention of how the models is validated.
Authors response: Thank you for noticing this. We have now added the suggested content. The revised section [lines 219-222] now reads as:
“After model training using the training set, model validation was carried out on the testing set (a new example dataset that was not used to build the model). During model validation, the model was assessed for underfitting, and overfitting the data. Necessary adjustment was carried out to enhance model accuracy/performance during validation phase.”
- It’s unclear whether the results shown are on the training datasets or testing datasets. The model performances must be shown on both datasets to evaluate if the models overfit or underfit.
Authors response: Thank you for brining attention to the clarity needed. We understand the importance of evaluating a model’s performance on both the training and testing datasets. The results shown are based on the testing dataset. To assess underfitting, and overfitting of the model, we used cross-validation. To run the cross-validation, we partitioned the data into multiple subsets, and we trained our model on a portion of the data while evaluating its performance on the remaining portion. we repeated this process multiple times and averaged the results to get an estimate of how well the model is likely to perform on new, unseen data [lines 210-214].
- There’s no mention of how the data leakage was prevented in the models.
Authors response: we appreciate this point and have included more detail in the method section explaining how the data leakage is prevented in the models [lines 166-171].
It reads as:
“We implemented various procedures in order to prevent data leakage. These include, dropping duplicate observation in the training and testing set, and removal of features that were highly correlated with the outcome variables such as the fasting plasma glucose (FPG) and hemoglobin A1C (HbA1C). We also did multicollinearity test between features using the variance inflation factor (VIF) test and removed variables that have a VIF value of greater than 5.”
- Figure 4: what’s the unit for x-axis.
Authors response: Thank you for noticing this. We have now added the unit of x-axis expressed in percentage (%) in the label [301-302].
- There’s no mention of whether any features were engineered to improve the model performance.
Authors response: Thank you for pointing this out. We have now revised the method section and added the steps we applied for feature engineering [lines 141-148].
The revised section now reads as:
“We undertook different steps of feature engineering including outlier detection, one hot coding, standardization of continuous variables, imputation to handle missing values and create one-hot matrix/encoding. The outlier data points were detected and removed using domain knowledge and data visualization. One-hot encoding was used to convert categorical variables into new variables that take on values 0 and 1 to represent the original categorical values. This format makes categorical variables to be used easily be machine learning algorithms. In addition, we imputed our data to handle missing values provided that the number of missing values of a given feature is less than 25%”.
- There’s no mention of how the models were tuned and optimized.
Authors response: We thank the reviewer for this excellent suggestion. As suggested, we have included the method on how the models were tuned and optimized, in the method section [lines 225-242]. The revised section now reads as:
“The tuneRF tool included in the RF Package was used for tuning the RF model. The RF model was tuned by using common hyperparameters including the number of trees used at the tuning step (ntree), number of variables randomly sampled as candidates at each split (mtry) and, node size. The optimization of the model was starting with the default value of hyperparameters and searching until the optimal hyperparameter value is obtained that gives smaller Out-of-Bag error estimate. The OOB error decrease as the number of trees increase which suggest improved performance of the RF model (eFig 1).
For the XGBoost model, learning curves were used for model tuning while Mlogloss of the training and testing set were applied as evaluation metrics. A learning curve is a plot that shows number of iterations in the x-axis and the mlogloss on the y-axis (eFig 2). The smaller the mlogloss indicates the higher learning or improvement in model performance. Two learning curves were drawn on the training ang testing sets. The curves were then used to diagnose under or overfitting of the XGBoost model. Accordingly, further hyperparameter tuning was carried out until the model became a good fit. This was achieved by tuning different hyperparameters including but not limited to the learning rate (eta), number of rounds (nrounds) and maximum tree depth (max.depth).”
- It’s difficult to justify what the data looks like. Please include the standard deviation for continuous variables in eTable1.
Authors response: Thank you for pointing this out. The standard deviation values were included for each continuous variable in eTable1.
Reviewer 2 Report
This paper employed multiple patient characteristics to efficiently predict uncontrolled diabetes mellitus using a supervised machine learning.
However, there are several comments to improve paper quality.
1. Please highlight the paper contribution on the introduction section to make easier for reader to pointing out the contribution.
2. In section 2.5, the reviewer cannot see the technique for class imbalance, whereas figure 1 shows hyperparameter tuning adjusting for class imbalance. Please explain it.
3. In fig 1, we see four steps for data processing and Machine-learning based Model Development steps in UDM using AoU Research Program data. It would be better if the subsections on materials and methods section followed for each step. So there are four subsections in this section. The goal of writing in this style is to make it easier for other people to follow the research method.
4. Please improve quality of figure.
Good luck.
Author Response
Dear Reviewer,
Thank you so much for your valuable comments. We have responded to your comments point by point as follows.
- Please highlight the paper contribution on the introduction section to make easier for reader to pointing out the contribution.
Authors response: Thank you for bringing attention to the clarity needed for the paper contribution in the introduction. We have added the suggested content [lines 95-103] and the revised section now reads as:
“The study highlighted the significance of physiologic biomarkers and patient characteristics to predict UDM in the absence of the regular glycemic status measurements. It is important to note that these clinical features are crucial in predicting UDM and identifying patients who require further management.
By utilizing a predictive model that incorporates various patient characteristics, healthcare providers can identify patients with UDM at an earlier stage, allowing for prompt intervention and management. Therefore, this paper aims to emphasize the significance of patient characteristics and biomarkers in predicting UDM, underline the importance of early intervention and prevention of complications associated with UDM.”
- In section 2.5, the reviewer cannot see the technique for class imbalance, whereas figure 1 shows hyperparameter tuning adjusting for class imbalance. Please explain it.
Authors response: We identified a misleading typographical issue regarding hyperparameter tuning and class imbalance in Graph 1. We solved it by putting “✔” sign to indicate that both procedures were done separately.
Hyperparameter tuning consists of finding a set of optimal hyperparameter values for a learning algorithm that maximizes the model's performance and reduce errors. However, hyperparameter tuning cannot handle class imbalance; instead, it has its own applications. [lines 225-226]
[lines 152-165]
“Handling imbalanced data involves preventing the effect of imbalanced values of the outcome variable on model performance. A data set with skewed proportions of the binary outcome variable is called imbalanced. In our study, 80% of patients had controlled DM while 20% had UDM, which gives an imbalance ratio of 1:4. Classes that make up a large proportion of the data set (controlled DM) are called majority classes, while those that make up a smaller proportion (UDM) are minority classes. We applied a random sampling technique using Random Over Sampling Examples (ROSE) package in R to resample the training dataset so that nearly equal proportions of UDM and controlled DM were created. We utilized three main approaches of random sampling. The first approach is called random under sampling, which deletes examples from the majority class, while the second approach, random oversampling, duplicates examples from the minority class. We also used a third approach by combining both random oversampling and under sampling techniques to determine which approach gives good learner.”
- In fig 1, we see four steps for data processing and Machine-learning based Model Development steps in UDM using AoU Research Program data. It would be better if the subsections on materials and methods section followed for each step. So, there are four subsections in this section. The goal of writing in this style is to make it easier for other people to follow the research method.
Authors response: Thank you for this insightful suggestion. In response to your suggestion, we have organized the subsections in methods to better reflect the flow of information presented in figure 1. Specifically, we have structured the methods section in the following order: the Source of Data and Population, Data Processing/Mining, Model Selection and Rationale, and Model Development and Optimization [Lines 172-175].
- Please improve quality of figure.
Authors response: We thank the reviewer for this valid point. We have now improved the quality of figures, especially figure 1 and 2, so that the contents in the diagram can be easily visualized.
Round 2
Reviewer 1 Report
Thank you for the revisions. This draft resolved the majority of the concerns from the previous review.
1. Please expand on the conclusion to make it more comprehensive.
2. Fig 4 looks photoshopped to me, please provide the original figure.
3. the two panels of fig 4 look exactly the same. however, in the previous version, these two panels are different for male and female models. Please clarify these are authentic and correct figures.
Reviewer 2 Report
Thank you for the revision.
The current version of this manuscript is much better than the previous version.
Some improvements are needed to accept completely.
1. Please add sentences between sections (x) and subsections (x.1, x.2, etc.).
2. Please refer to the citation style based on the MDPI format. The current version is different from the MDPI reference format.
3. Conclusion is too short. From four sentences, it contains three sentences for the conclusion and one sentence for limitation. Please emphasize it with more structure (conclusion and future work).
good luck
